# Terahertz-field activation of polar skyrons

Huaiyu Hugo Wang [1,2,18], Vladimir A. Stoica [1,3,18], Cheng Dai[1,18], Marek Paściak [4,18], Sujit Das [5], Tiannan Yang[1,17], Mauro A. P. Gonçalves [4], Jiri Kulda[6], Margaret R. McCarter[7], Anudeep Mangu[8], Yue Cao [9], Hari Padma [1], Utkarsh Saha[1], Diling Zhu[10], Takahiro Sato [10], Sanghoon Song [11], Matthias C. Hoffmann [11], Patrick Kramer [11], Silke Nelson[11], Yanwen Sun [11], Quynh Nguyen[11], Zhan Zhang [3], Ramamoorthy Ramesh[7,11,12,13,14,15], Lane W. Martin [11,13,14,15,16], Aaron M. Lindenberg [2,8], Long-Qing Chen [1], John W. Freeland [3] ✉, Jirka Hlinka [4] ✉, Venkatraman Gopalan [1] ✉ & Haidan Wen [3,9] ✉

Unraveling collective modes arising from coupled degrees of freedom is crucial for understanding complex interactions in solids and developing new functionalities. Unique collective behaviors emerge when two degrees of freedom, ordered on distinct length scales, interact. Polar skyrmions, three-dimensional electric polarization textures in ferroelectric superlattices, disrupt the lattice continuity at the nanometer scale with nontrivial topology, leading to previously unexplored collective modes. Here, using terahertz-field excitation and femtosecond x-ray diffraction, we discover subterahertz collective modes, dubbed "skyrons", which appear as swirling patterns of atomic displacements functioning as atomic-scale gearsets. The key to activating skyrons is the use of the THz field that couples primarily to skyrmion domain walls. Momentum-resolved time-domain measurements of diffuse scattering reveal an avoided crossing in the dispersion relation of skyrons. Atomistic simulations and dynamical phase-field modeling provide microscopic insights into the three-dimensional crystallographic and polarization dynamics. The amplitude and dispersion of skyrons are demonstrated to be controlled by sample temperature and electric-field bias. The discovery of skyrons and their coupling with terahertz fields opens avenues for ultrafast control of topological polar structures.

Spontaneous symmetry breaking involving lattice, spin, or charge degrees of freedom in crystalline materials can give rise to ordered phases such as charge-density and spin-density waves, and superconductivity, profoundly influencing their electronic, magnetic and thermal properties[1]. Besides spontaneous symmetry breaking, engineering precisely ordered superstructures can break additional symmetry on different length scales. These engineered orders interact with intrinsic ones to generate hybrid collective excitations with extraordinary properties. For example, twisting van der Waals heterostructures gives rise to moiré superlattices that interact with Bloch waves of electrons, resulting in tunable superconductivity[2–4]. Precisely stacked atomic layers in oxides balance the competing equilibrium states and reshape the energetic landscape, enabling emergent order and properties that couple with the lattice[5]. Creation, characterization, and control of hybrid modes arising from symmetry breaking at distinct length scales are crucial to advance the understanding of many-body physics and to inspire next-generation technologies[6,7].

In ferroelectric materials, spontaneous polarization arising from ionic displacements in the unit cell breaks inversion symmetry. Using electric polarizations as building blocks, distinctive nanometer-scale

polar structures can be engineered and imposed on the existing crystalline order in ferroelectric superlattices[8], including polar vortices[9,10], skyrmions[11], merons[12], hopfions[13], and light-induced supercrystals[14–16]. Among this family of polar nanostructures, polar skyrmions are three-dimensional topological objects hosted in a $[(PbTiO_3)_{16}/(SrTiO_3)_{16}]_8$ superlattice (SL) grown on $SrTiO_3$ (001) substrates[11]. The polarization of adjacent unit cells reconfigures to form "nanobubbles" with nontrivial topology on the order of a few nanometers (Fig. 1a, insert), offering a testbed for electric-field switchable and tunable ferroelectric chirality[17]. Unlike magnetic skyrmions, the topological configurations of polarization are strongly coupled to the lattice and can create elastic waves with spin-like properties[18,19]. The dynamics of polar skyrmions are not yet known, in particular, as a result of strong coupling of nanoscale skyrmion order with the atomic-scale polarization order. The conventional collective excitations, such as elastic waves and polarization waves, known as ferrons[20,21], may hybridize into collective modes unique to polar skyrmions. Previous investigations of stimulated polar-skyrmion dynamics were limited to theoretical modeling[22,23] and experiments conducted under quasi-equilibrium conditions[24,25]; their non-equilibrium dynamics and collective excitation have just started to be explored. Understanding the dynamics of polar skyrmions is an essential prerequisite for their potential applications in data storage, data processing, and polaritronic devices[26,27].

## Results

Using the ultrafast THz-pump, femtosecond X-ray diffraction probe technique (Fig. 1a), aided by theoretical modeling, we reveal the dynamical properties of the polar skyrmions. Complementary to second-harmonic-generation probes[28], the momentum-resolved measurements provided by ultrafast x-ray diffraction are directly sensitive to the order parameter of polar skyrmions, which are critical to reveal collective atomic motions and their unique dispersion. Four satellite peaks around the specular SL rod arise from the quasi-four-fold symmetry of the in-plane skyrmions ordering. The Ewald sphere cuts through the center of the SL peak as well as two satellite peaks, allowing for a simultaneous recording of their diffraction by an area detector (Fig. 1b). Representative differential patterns between the excited and the equilibrium states are plotted as a function of time (Fig. 1c). Upon THz excitation at time zero, which is defined as the arrival time of the peak of the THz field, modulation of the diffraction intensity was observed in the satellite peaks (Fig. 1c). Detailed analysis of such modulation reveal that the confinement of polarizations in nanobubbles creates collective atomic motions acting as three-dimensionally integrated atomic gearset, unlike quasi-two-dimensional dynamics observed in polar vortices[29] or conventional ferroelectrics[30]. In addition, the diffuse satellite scattering offers the opportunity to examine the dispersion relation of skyrmion collective modes, revealing avoided crossing of phonon bands as a result of strong hybridization of polarization and elastic waves, which is absent in magnetic skyrmions[31].

## Collective dynamics of polar skyrons

To quantitatively assess the dynamics of the satellite diffraction peaks, we track the intensities of the satellite peaks across two mini-Brillouin zones as a function of time (Fig. 2a). The mini-Brillouin zone refers to a region in the reciprocal space with a size set by the inverse of the periodicity of the polar-skyrmion lattice. By analyzing the intensity oscillation at representative $Q_x$ along the vertical dashed lines in Fig. 2a, it was found that the amplitude of the oscillation peaks at the maximum of the driven THz field, similar to observations in driving soft phonon modes[32,33] or electromagnon[34]. After the THz pulse exits the probe volume in the sample, the resulting oscillation deviates from the shape of the THz pulse, indicating an intrinsic sample response.

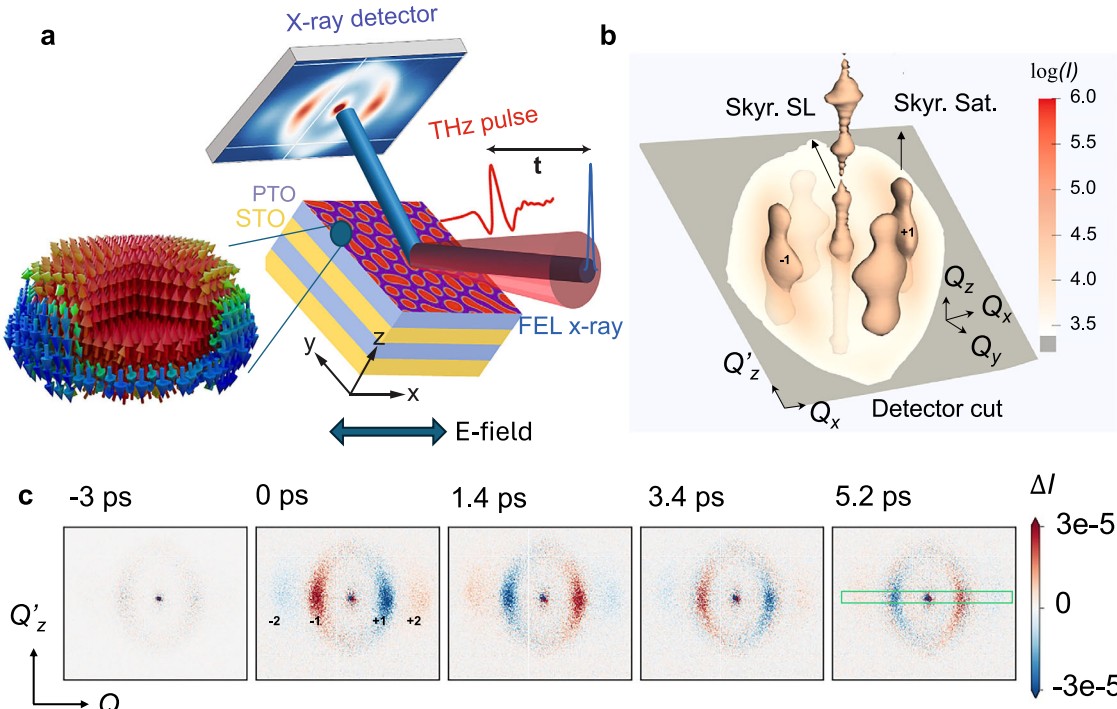

**Fig. 1 | Probing dynamics of polar skyrmions by intense THz excitation and femtosecond X-ray diffraction. a** Experimental setup of THz pump X-ray diffraction probe experiment using an X-ray free-electron laser (FEL). A zoom-in view of the bubbles in the $[(PbTiO_3)_{16}/(SrTiO_3)_{16}]_8$ SL shows the structure of a polar skyrmion, with the red and blue arrows representing up and down polarizations, respectively. **b** Illustration of the reciprocal space mapping of polar skyrmions with the $Q_x$-$Q_z'$ detector cut overlaid. **c** Differential detector images near 004 Bragg peak at the representative time delays, showing the change in diffraction intensity ($\Delta I$). The integer number labels the corresponding orders. The green box illustrates the region of interest for time domain analysis.

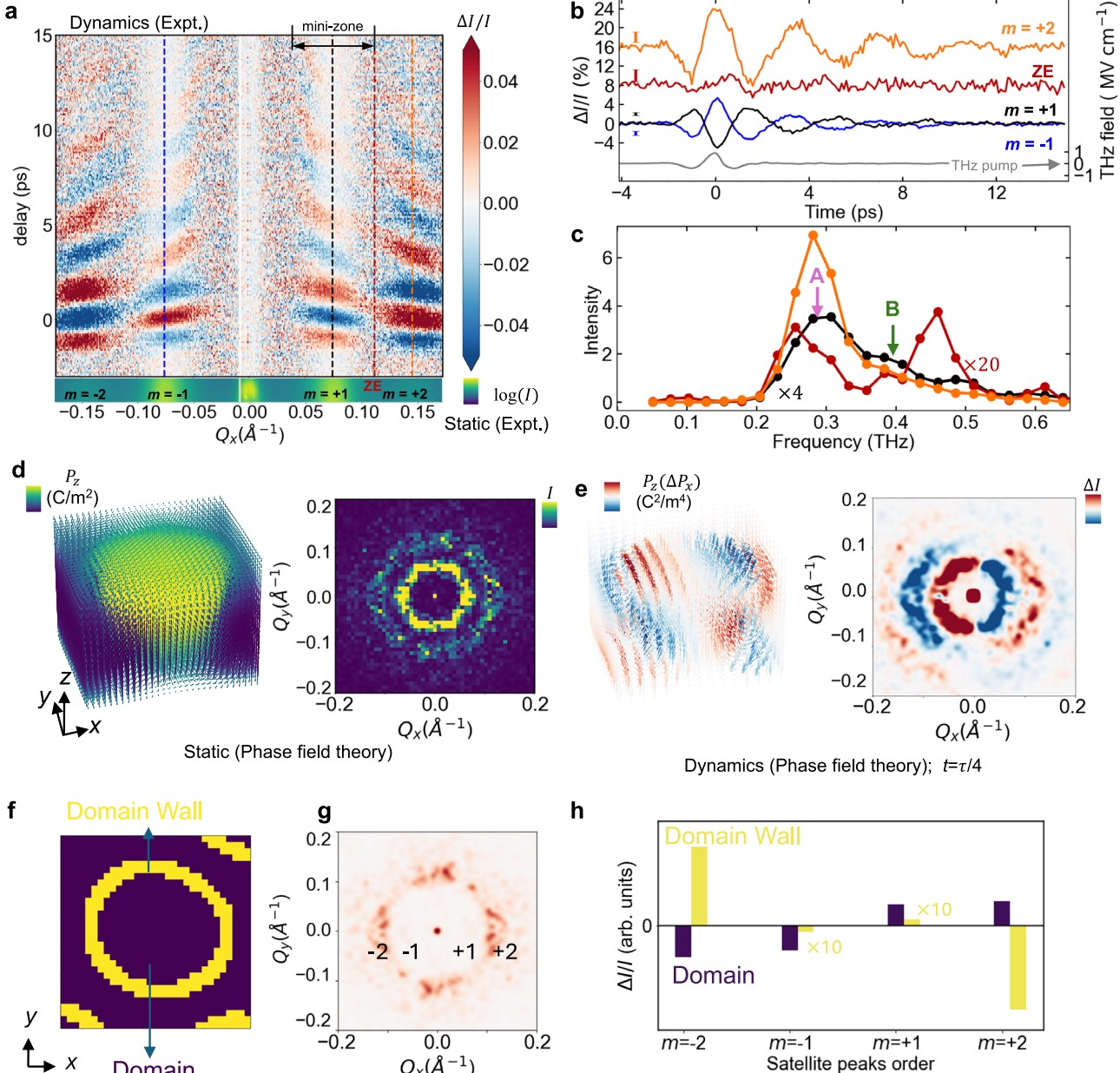

**Fig. 2 | Dynamics of polar skyrons. a** Region of interest indicated by the rectangular box in Fig. 1c is integrated along the $Q_z$ direction near 013 peak. The change of intensity ($\Delta I$) is normalized by the diffraction intensity ($I$) before time zero is plotted as a function of $Q_x$ and the delay. m: the order number of the satellite peaks. ZE: zone edge. **b** The time-dependent diffraction intensity changes at the selected $Q_x$ as indicated by the same color-coded dashed lines in (**a**). **c** Fourier spectra of the time-dependent diffraction intensity in (**b**). The error bar represents the standard deviation of the data points before time zero. The frequencies of modes A and B discussed in the text are indicated by the arrows. **d** Static polar Skyrmion structure in real space obtained by phase-field simulation and calculated diffraction intensity of the static structure in reciprocal space. **e** Dynamical phase-field simulation of a snapshot of the perturbed polar skyrmion structure with the maximum polarization change. The arrows indicate polarization vectors of the unit cells. Their colors represent the magnitude of $P_z\Delta P_x$. The right panel shows the corresponding diffraction intensity changes in the reciprocal space. **f** Schematic of domain and domain wall regions in a polar skyrmion. **g** Simulated diffraction intensity of the domain walls that mainly contributes to m = ± 2 satellite peaks. **h** Analytical calculation of dynamical responses that separate the contribution from domain and domain walls at the maximum polarization change (See Supplementary Text 1).

The Fourier transformation of the time-domain data reveals a dominant frequency at 0.27 THz (henceforth, mode A) and a secondary frequency at 0.39 THz (henceforth, mode B) (Fig. 2c). These modes are dubbed "skyrons" due to their unique dynamics and dispersion, which will be detailed later. Close to the mini-zone edge (dark red), two spectral peaks are visible at 0.25 THz and 0.46 THz, which belong to the dispersion curves of modes A and B, respectively.

We first focus on the observation of the intensity oscillation of the polar skyrmions satellite peaks (Figs. 1c, and 2b). It was found that the pair of m = 1 and −1 satellite peaks (Fig. 2b) oscillate out of phase after

THz excitation; this opposite phase is also observed between the m = 2 and −2 peaks. Furthermore, the phases between m = 1 and 2 peaks are also opposite, as it is between m = −1 and −2 peaks. These observations contrast the same phase oscillation observed in polar vortices (Fig. S1c, d), as well as in the case of optical excitation of polar skyrmions (Fig. S9). The intensity oscillation can be reproduced by dynamical phase-field calculations[35], in which the polar skyrmion structure is perturbed with the same THz waveform as in the experiment (see "Methods"). After obtaining the static skyrmions and their diffraction (Fig. 2d), we computed the perturbed structures as a function of time

following the THz field. The modulation of the skyrmion supercells leads to transient nanostructures that function as a blazed grating at hard X-ray frequency[36] to dynamically redistribute the diffraction intensity from +Q and -Q satellite peaks (Fig. S2c), resulting in the opposite-phase intensity change between $m = \pm 1$ and $\pm 2$ peaks in agreement with the simulation (Fig. 2e and Fig. S2a). In contrast, similar simulations for the polar vortices show the same phase oscillation upon THz excitation (Fig. S2b), which agrees with the experimental results (Fig. S1b). Through a combined approach of phase-field simulations and analytical approximations (Supplementary Text 1 and Figs. S2, S3), it was found that the main contributor to opposite-phase oscillation in diffraction intensity is the polarization dynamics described by the term of $P_z\Delta P_x$, whose spatial distribution further breaks the mirror-plane (y-z plane across the center of the skyrmions) symmetry (Fig. 2e), in addition to the symmetry breaking by the Bloch-domain wall around the equator of polar skymions. Furthermore, phase-field diffraction simulations show that the $m = 2$ diffraction peak is predominantly sensitive to the domain wall of polar skyrmions (Fig. 2f, g). The out-of-phase dynamics observed between the first- and second-order satellite peaks arise from the distinct dynamical responses of the domain walls and domains (Fig. 2h).

## Dispersion and modeling of polar skyrons

The intensity spread of the satellite diffraction peaks almost covers the whole mini-Brillouin zone along $Q_x$, offering the opportunity to study $Q_x$-dependent intensity oscillations (Fig. 2a). The Fourier transform of the intensity oscillation reveals the dispersion of the polar skyrmion modes (left side, Fig. 3a). The dispersion of these modes provides direct evidence of the modified energy transport at nanoscale via acoustic phonons, different from the conventional transverse acoustic (TA) waves. To guide the eye, the central frequencies of the $Q_x$-dependent Fourier components are illustrated by the dashed curves, revealing two distinct branches. At the mini-Brillouin zone centers ($\pm 0.075 \,\text{Å}^{-1}$), the two branches exhibit avoided crossing: branch A bends down at higher $|Q_x|$ while branch B curves up at lower $|Q_x|$. Both branches deviate from the linear TA rather than LA dispersion, indicating stronger renormalization with TA phonons. The avoided crossing is phenomenologically similar to that of a ferron polariton[20]; however, the regime of the observed dispersion relation is far away from the photon-dispersion curve (the solid gray line marked by "P"). Instead, this avoided crossing of the phononic dispersion indicates strong renormalization of phonon modes. This is because skyrmions break the discrete symmetry of the lattice by forming ferroelectric domains and domain walls. Therefore, the conventional TA dispersion is interrupted at the wave vector of skyrmions due to the acoustic impedance mismatch within and outside of skyrmion bubbles. Although the phonon renormalization has been observed in heterostructures of dissimilar materials[37], it is remarkable that such mismatch can occur in a single phase, i.e., PbTiO$_3$ layer in our case, demonstrating phononic engineering in single-phase materials.

The dispersion map of the polar skyrmions was compared with the results around the 013-diffraction peak obtained using atomistic first-principle-based molecular-dynamics simulations (Methods) (right side, Fig. 3a). Two branches shown by the calculation intersect the mini-zone center of the $m = 1$ peak at the frequencies of 0.29 and 0.35 THz. Without any rescaling, these calculation results agree with the observed frequencies of 0.27 and 0.39 THz shown by the red dots in Fig. 3a for mode A and B within the experimental error bar, respectively. The calculated dispersion also shows strong deviation from the conventional TA branch at the mini-zone center. The slope of the dispersion at the mini-zone center, quantifying the group velocity of the acoustic wave, is two times lower than the conventional TA speed, indicating the modification of phonon transport at this $Q_x$ due to the disruption of polar skyrmions. The agreement between experimental results and simulation is also evident around the 004-Bragg

peaks (Fig. S4). Despite different Bragg peaks being probed, the time-domain responses and dispersions remain similar, demonstrating the generality of the modified phonon dispersion.

The atomistic calculation reveals the microscopic polarization and atomic dynamics of modes A and B (Supplementary movies 1–4). Mode A features opposing vector fields of the polarization changing along the x-axis between the top and bottom of the skyrmion (Fig. 3b and Fig. S5a), which leads to shear strain as shown by the lead (Pb) displacement in the x-z plane (Fig. S6a). The cross-sectional view in the x-z plane (C1) reveals two vortexons, i.e., atomic swirling patterns of Pb displacements[29], which are stacked vertically (Fig. 3b). The horizontal cut in the x-y plane (C2) shows a dominant Pb displacement along the x-axis. Mode B features polarization changes that divide the skyrmion into the four quadrants of the skyrmion (Fig. 3c and Fig. S5b). The Pb displacement map shows four vortexon-like patterns with alternating rotation directions (Fig. 3c). The vertical cut on the side of the skyrmion (C3) shows a representative pattern. The top cut in the x-y plane (C4) shows an anti-vortex pattern, and other cuts can be found in Fig. S6b, which are consistent with the integrated gearset illustrated in the middle of Fig. 3c.

The localized polarization dynamics of these modes lead to localized shear strain across the supercell of the polar skyrmions (Fig. S7a). To calibrate the magnitude of the shear strain, we simulated the diffraction change based on the atomistic model (Supplementary Text 2). The best match of the simulation with the experimental data gives an estimate of the mean shear strain ($\varepsilon_{xz}$) up to -1% for mode A (Fig. S7b). For an applied THz peak field of 535 kV cm$^{-1}$, which has considered the footprint of the THz beam on the sample, this shear strain corresponds to a piezoelectric coefficient d$_{15}$ of 187 pC N$^{-1}$, which is larger than the d$_{15}$ of 56.1 pC N$^{-1}$ in PbTiO$_3$[38]. These observations open opportunities for designing strong piezoelectric materials at sub-terahertz frequencies as well as dynamically manipulating polar topology[23].

## Control of polar skyrons

The dynamic response in the polar skyrmions can be controlled via temperature as well as an external electrical bias. As the temperature increased beyond ~360 K, the magnitude of the dynamic response was significantly reduced (Fig. 4a and Fig. S8b). At 380 K, the signature of avoided crossing is significantly reduced (Fig. 4b). A similar control was observed when applying an in-plane DC bias with a field of 25 kV cm$^{-1}$ to the polar skyrmions (Methods), shown by the red star in Fig. 4a (data shown in Fig. S8c). The temperature and field dependence of the dynamical response can be ascribed to a change in morphology from skyrmion bubbles at 300 K to a labyrinth state at 360 K, revealed by the phase-field simulations (Fig. 4c). A similar transition from skyrmion bubbles to the labyrinth structure via in-plane DC electric bias has also been predicted[39]. This transition is evident by the increase in static diffraction intensity of the first-order satellite peaks, accompanying a peak width narrowing (Fig. S8a). The weakened avoided crossing in the labyrinth phase can be attributed to changes in domain wall morphology. As skyrmion bubbles evolve into stripe-like labyrinth domains, their size increases along one dimension, allowing acoustic waves to propagate more freely in that direction, exhibiting the unaltered TA dispersion. However, since the domains remain confined in the orthogonal directions, the avoided crossing is reduced but not fully eliminated. The temperature and in-plane DC field-dependent dynamical responses demonstrate the tunability of the dynamics of polar skyrmions, which originates in the intricate competing phases in topological polar nanotextures.

The temperature-dependent amplitudes of the coherent oscillations at $m = \pm 2$ diffraction are different from those of the $m = \pm 1$ diffraction (Fig. 4c). The amplitude of the $m = \pm 2$ diffraction intensity oscillation reaches a maximum at 320 K, while that of the $m = \pm 1$ diffraction does not. This can be explained as the $m = \pm 2$ diffraction is

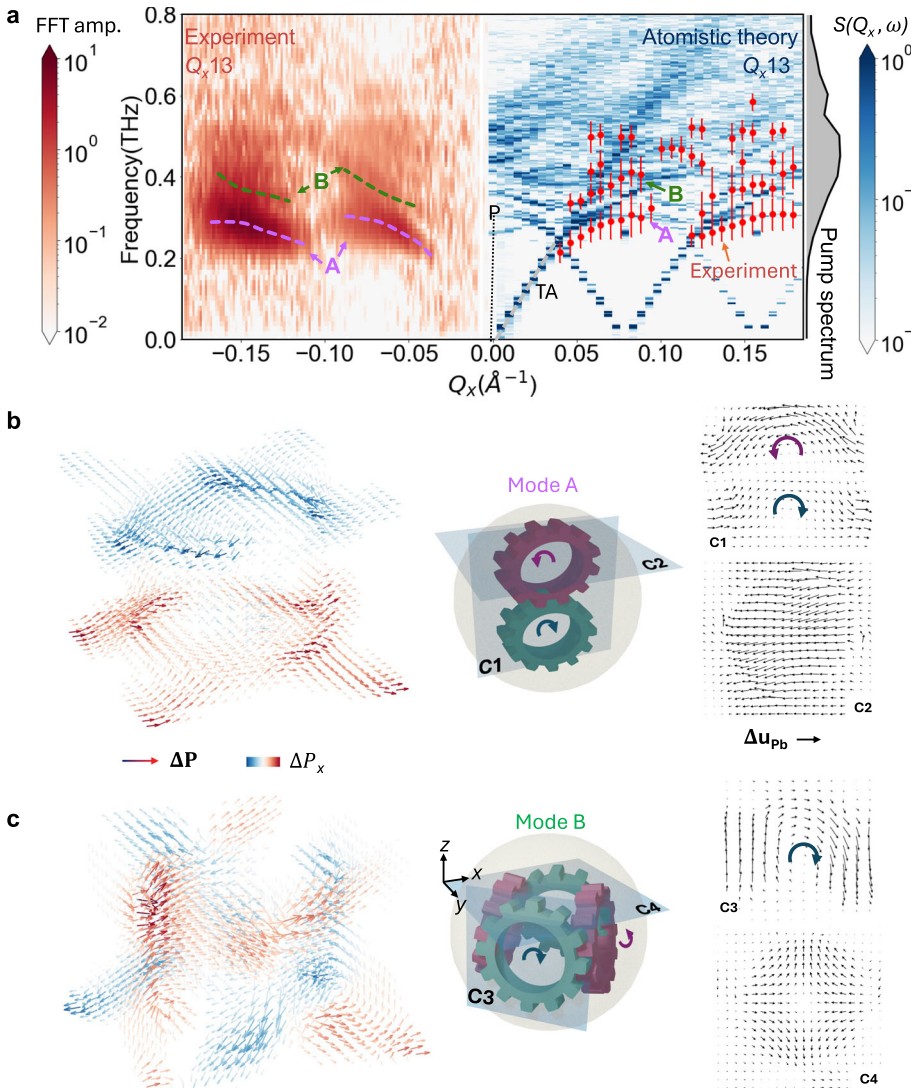

**Fig. 3 | Dispersion relation of polar skyrons and their microscopic dynamics.** **a** Fourier spectra of Fig. 2a (left), compared with the results by the atomistic simulation (right). The color map of the simulation represents the amplitude of the dynamic structure factor $S(Q_x, \omega)$. Red dots are the overlaid experimental data with the error bars that show the FWHM of the Lorentzian peak fitting of Fourier spectra at each $Q_x$. Two dispersion branches are labeled as A and B mode. The dispersion outside of the THz-pump spectrum shown on the right axis is not experimentally discernible. The dispersions of polariton (P), transverse acoustic (TA) phonons are shown by the dashed lines for comparison. **b, c** Snapshots of mode A and B at the maximum of the oscillation amplitude, respectively. The left column shows the change of polarization vectors (colored arrows). The middle column shows the schematics of skyrons, in which the dynamic vortices within the skyrmions are represented by the rotating gears. The right column shows the change of Pb displacement in the planes as indicated by labels C1 to C4 in the middle schematics. The thick curved arrows are guide of eye for visualizing the dominant dynamics of the respective regions.

more sensitive to the domain walls. During the transition from skyrmion bubble to labyrinth structure, domain walls significantly reconfigure close to the transition temperature, leading to their enhanced susceptibility to the THz excitation and thus the increase of the oscillation amplitude at 320 K. Once the labyrinth structures dominate, domain wall density is significantly reduced, so as the dynamical responses. The correlation between domain wall density and the oscillation amplitude highlights the leading role of domain walls in driving the dynamical response.

## Discussion

THz-field excitation is critical to revealing the skyron modes discussed above because the THz fields selectively couple to ionic displacement[29,32,33,40–43], rather than exciting electronic transitions. To show the distinct dynamics upon electronic excitation, optical pump x-ray diffraction probe measurements were performed around the 004-Bragg peak, where the pump photon energy of 3.1 eV is

above the band gap of $PbTiO_3$. The optical excitation results in different dynamics from those excited by the THz field. First, the intensity of the SL Bragg peaks upon the 400 nm optical pump decreases (Fig. S9a). This is due to the strain effect that shifts the Bragg peak along the $Q_z$ axis, indicating a strong excitation of out-of-plane strain waves. Second, satellite intensity oscillations have the same phase after subtracting the step function and the exponential decay (Fig. S9b). This is because the stress induced by the optical excitation is mainly along the out-of-plane direction and does not further break the mirror symmetry of the sample in the in-plane direction as the THz field does. Third, the Fourier spectrum of background-subtracted data shows a linear dispersion relation, whose slope is consistent with the speed of sound of the TA waves in $PbTiO_3$ along the [100] axis[44] (Fig. S9c). The deviation from the linear dispersion as observed upon THz excitation (Fig. S9d), especially near the first-order satellite peak center, suggests that the in-plane THz-field excitation stimulates a set of hybridized polarization

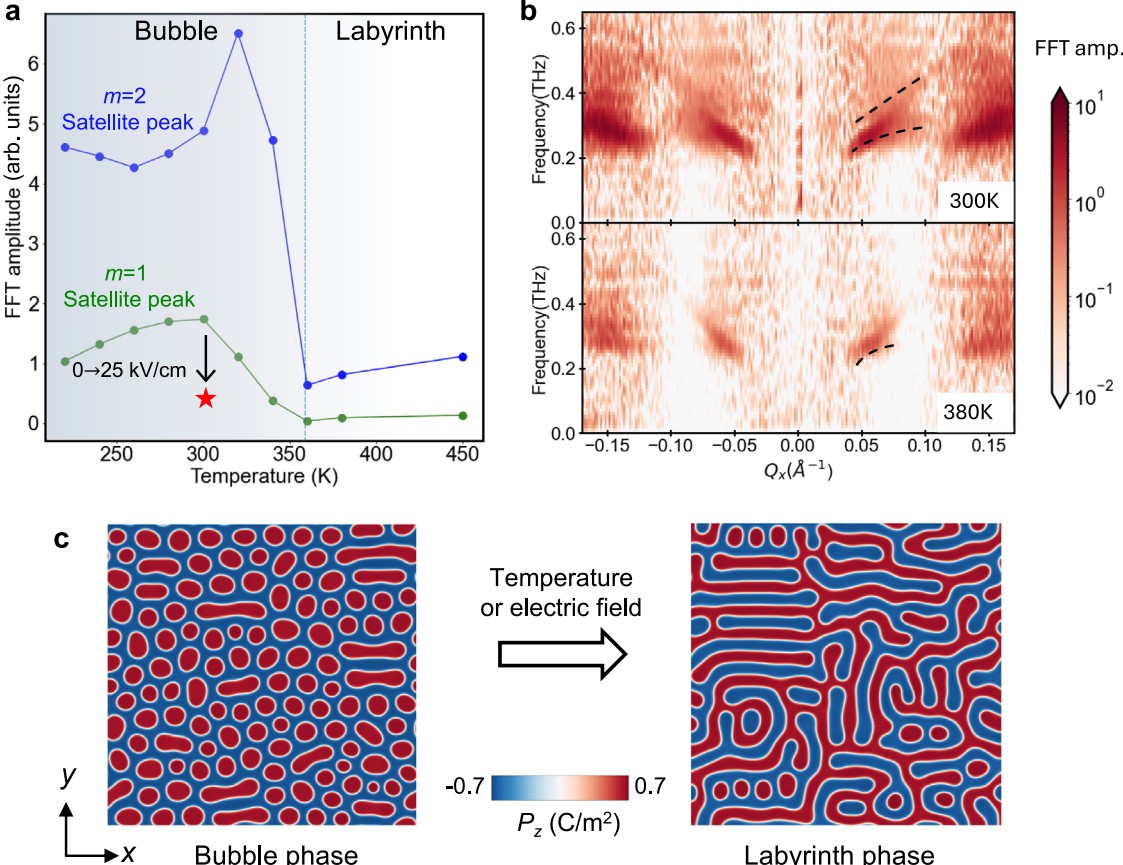

**Fig. 4 | Control of polar skyrons by temperature and in-plane electric-field bias.** **a** Fourier amplitude of the $\Delta l//l$ oscillation near the 004 peak measured at the first (green) and second (blue) orders of the skyrmion satellite peaks as a function of temperature. The in-plane DC bias field can reduce the Fourier amplitude, as indicated by the black arrow (data shown in Fig. S8c). The error bar is smaller than the marker size. The dashed line indicate the transition temperature from The skyrmion bubble to labyrinth phase. **b** Dispersion relation measured near 004 peak at (top) 300 K, and (below) 380 K. The dashed black curves are the guides for the eye. **c** Phase-field simulation captures a phase transition from a stable bubble phase at 300 K to a vortex-tube-like labyrinth phase at 360 K. The color-map indicates the amplitude of $P_z$.

modes rather than the conventional acoustic response. Finally, a pronounced dynamical response of $m = \pm 2$ diffraction was observed only under THz excitation, but not discernible upon the 400 nm optical excitation. This highlights that the THz pulse strongly couples to the domain walls, whereas optical excitation primarily couples to the domains but does not activate the domain walls.

The combined experimental and theoretical investigations clarify the overall physical picture of how a THz field activates polar skyrons - modes originating from the unique polarization configuration of polar skyrmions. Although the THz field interacts with the dipole moment of every unit cell, the response at skyrmion walls is particularly strong, due to the polarization gradient leading to highly susceptible walls. The excitation of domain walls activates acoustic waves which hybridize with the polarization waves. This hybridization, manifesting as a characteristic avoided crossing of phonon branches in the sub-THz range, results in the collective atomic motions in the form of dynamical swirling patterns. More broadly, skyrons are intimately related to the physics of collective modes arising in charge-density-wave systems. Polar skyrmions are incommensurate superstructures that break the discrete translation symmetry rather than the U(1) phase symmetry. Similar to charge density waves, their locations are not necessarily pinned and may translate, which gives rise to phason-like excitation[45]. When the THz-field interacts with polar skyrmions, the dynamical response does not lead to the displacement of polar skyrmions but can stimulate localized structural distortion predominantly at the domain walls, which can be regarded as the excitation of inhomogeneous phasons[46].

In conclusion, ultrafast dynamics of skyrons, i.e., collective modes of polar skyrmions, were directly measured by THz-pumped femtosecond x-ray diffraction, revealing the avoided crossing of the acoustic phonon band in a low-Q and low-energy regime that is difficult to access previously. This key observation of the dispersion relation depicts an essential characteristic of polar skyrmions that significantly modifies nanoscale energy transport. The dispersion relation of skyrons, their dependence on the sample temperature and electric bias, as well as the measurement of a strong piezoelectric response at sub-terahertz frequencies, pave the way for dynamical control of topological polar textures for ultrafast and ultradense ferroelectric-based devices.

## Methods

### THz-pump, ultrafast x-ray diffraction experiments

The experiments were carried out at the X-ray pump probe beamline of the Linac Coherent Light Source (LCLS) at SLAC National Laboratory[47]. The LCLS provides linearly polarized pulsed X-ray of 40 fs pulse duration at 120 Hz. Hard X-ray pulses at 9.8 keV were selected by a monochromator and focused to $200 \times 200\ \mu m^2$ beam size by Beryllium lenses. The sample was mounted on a goniometer in a horizontal scattering geometry. The temperature control was achieved by a cryojet that provides a steady stream of cold nitrogen to reach cryogenic temperature. The high-temperature data was collected with a heater in the sample holder. A temperature sensor was mounted beneath the sample to record sample temperatures. A

scattered X-ray diffraction beam was recorded shot-by-shot using an area detector (Jungfrau 1 M). The in-plane electric field bias was applied via interdigital electrodes deposited on the sample with a gap of 8 μm.

A Ti:sapphire pulse laser synchronized to the free electron laser was used to generate 100-fs, 800nm laser pulses with around 20 mJ pulse energy, which were then converted to single-cycle terahertz pulses by the pulse-front-tilt method in a LiNbO3 prism[48]. The THz pulses were vertically polarized with a maximum peak electric field of ~800 kV cm$^{-1}$, as calibrated by electro-optical sampling measurement using a 50 μm-thick GaP crystal. The THz beam was focused on the sample collinearly with the X-ray beam using a parabolic mirror. In the optical pump setup, 400 nm pulses were generated via the second harmonic generation process in a 100 μm thick $\beta$-BaB$_2$O$_4$ crystal. The incident fluence of 6.5 mJ cm$^{-2}$ of a 400 nm pump beam was used to excite the sample. The spatiotemporal overlap between optical and X-ray was achieved using the 't0-finder'[49].

X-ray diffraction patterns around the specified group of Bragg peaks were acquired as a function of the delay between THz/optical pump and X-ray probe pulses. The time jitters in X-ray pulses were compensated by the 'timing tool'[50] to reach a temporal resolution of around 50 fs. The delay scan adopted a continuous back-and-forth mechanical delay stage motion to improve the data collection speed[47]. The shot-by-shot detector images were grouped into laser-on and laser-off data sets. The detector signal was normalized to the intensity monitor I$_0$, which is proportional to the X-ray photons in each pulse. The signal was selected based on I$_0$ ranges where the correlation between diffraction intensity on the detector and I$_0$ is linear. Typically, 600–1000 pump-probe events were accumulated for each data point, which was grouped into 100–200 fs temporal bins as needed.

### Sample preparation

The [(PbTiO$_3$)$_{16}$/(SrTiO$_3$)$_{16}$]$_8$ superlattice samples on SrTiO$_3$ substrate were fabricated using reflection high-energy electron-diffraction-assisted pulsed laser deposition[11]. Structural characterization and preliminary time-resolved measurements of the samples were carried out using synchrotron-based X-ray diffraction at beamlines 33-ID-C and 7-ID-C of the Advanced Photon Source, respectively. Three-dimensional reciprocal space maps characterized the structural properties of polar skyrmion structures in the superlattices.

### Phase-field simulations

We employed the dynamical phase-field model (DPFM) to simulate the polarization **P** and mechanical displacement **u** evolution during optical excitation. The evolution of **P** and **u** are governed by the polarization dynamics equation and the elastodynamics equation, respectively.

$$\mu \frac{\partial^2 \mathbf{P}}{\partial t^2} + \gamma \frac{\partial \mathbf{P}}{\partial t} + \frac{\delta F}{\delta \mathbf{P}} = 0 \tag{1}$$

$$F = F_{Landau} + F_{elastic} + F_{electric} + F_{Gradient} \tag{2}$$

$$\rho \frac{\partial^2 \mathbf{u}}{\partial t^2} = \nabla \left( \boldsymbol{\sigma} + \beta \frac{\partial \boldsymbol{\sigma}}{\partial t} \right) \tag{3}$$

In Eq. (1) (polarization dynamic equation), $\mu$ and $\gamma$ are the mass and damping coefficient of the polarization, and $F$ is the total free energy of the system with the expression as Eq. (2). The time-dependent THz field $E(t)$ modifies the free energy term $F_{electric} \propto E(t)P(t)$. In Eq. (3) (elastodynamics equation), $\rho$ and $\beta$ are the material mass density and the elastic stiffness damping coefficient of the material, and $\boldsymbol{\sigma}$ represents the stress field. Energy component and numerical solution about the DPFM are followed by literature[29,51,52]. The diffraction pattern derived from the phase-field simulation results is

calculated as following[35]:

$$I(\mathbf{q}) \propto |F(\mathbf{q})|^2 \tag{4}$$

$$F(\mathbf{q}) = \sum_{n,m,l} f_{n,m}(\mathbf{q}) \eta_{m,l} e^{-i\mathbf{q} \cdot \mathbf{R}_{n,m,l}} \tag{5}$$

$$\mathbf{R}_{n,m,l} = \mathbf{R}_l + \Delta \mathbf{R}_n + \mathbf{u}_{n,m}(\mathbf{R}_l) \tag{6}$$

$$\mathbf{u}_{n,m}(\mathbf{r}) = \mathbf{u}(\mathbf{r}) + \sum_h b_{n,m,i} P_i(\mathbf{r}) \tag{7}$$

where $I$, $\mathbf{q}$ and $F$ are the scattering intensity, wave vector in the reciprocal space and the structural factor, respectively. $\mathbf{R}_{n,m,l}$ is the atomic position. where $\mathbf{R}_l$, $\Delta \mathbf{R}_n$ and the $\mathbf{u}_{n,m}(\mathbf{r})$ are the position vectors of the l-th unit cell, the relative position of the n-th atom and the mechanical displacement of the n-th site in the m-th phase, respectively. $b_{n,m,i}$ is the tensor that measures the dependence between the electrical polarization $P_i(\mathbf{r})$ and the atom position of the n-th site in the m-th phase, which can be obtained from DFT calculations. More details about the diffraction pattern calculation are described in previous literature[35].

(SrTiO$_3$)$_{16}$/ (PbTiO$_3$)$_{16}$/(SrTiO$_3$)$_{16}$ slab is discretized with a mesh of 200 × 200 × 48 grids, in which each grid represents 0.4 nm. A three-dimensional periodic boundary condition is employed, for the mechanical boundary condition, the in-plane directions are clamped while the out-of-plane is assumed to be stress-free. All constants are listed in Table S1.

### Atomistic simulations

We exploit two methods of accessing the system dynamics on the fully atomistic level. First is based on classical analysis in which the dynamical matrix is built and diagonalized, yielding frequencies and eigenvectors of the system's normal modes. The second is oriented towards the analysis of a finite-temperature dynamical structure factor $S(Q_x, \omega)$ as obtained through the processing of a molecular dynamics trajectory. The two approaches are complementary since the dynamical matrix analysis gives precise information in the real space, while $S(Q_x, \omega)$ provides insights into the reciprocal space dispersion curves. The agreement of the simulation results with the experimental dispersion is achieved without any rescaling either along the momentum or frequency axis, demonstrating the reliability of the simulation results.

A prerequisite for the application of both analytical methods is the possibility of calculating forces for the superlattice system. Since polar skyrmions are mesoscale objects, quantum mechanical, density functional theory-based calculations are not available and ab-initio based, but classical interatomic potentials are a computationally affordable choice. Here we use shell-model potentials that have been especially suited for dielectric materials, with the particular parametrization for PbTiO$_3$ taken from Sepliarsky and Cohen[53]. SrTiO$_3$ parameters are that of Li et al.[29], where both sets of parameters were already used for the PTO/STO superlattice, successfully describing the structure and dynamics of the vortex-tube system in the PTO layer.

Guided by the sample's superlattice period and experimental observations for the sizes of skyrmions, the elementary simulation box was set to comprise (PbTiO$_3$)$_{16}$/(SrTiO$_3$)$_8$ layers with in-plane dimensions of 20 × 20 perovskite unit cells. A thinner STO layer has been chosen to reduce computational effort since the skyrmion bubble inhabits only the PTO layer. The initial configuration of the PTO consisted of a cylindrical domain with a radius of 6 unit cells embedded within a matrix of inverted polarization with an additional neutral 180-degree domain wall region of one unit cell thickness.

The simulation box was pre-optimized with the steepest-descent procedure using the molecular dynamics package DL_POLY3[54]. Subsequently, the program GULP[55] has been used for a Newton-Raphson optimization and calculation of system properties including Born effective charges, phonon frequencies and eigenvectors (from dynamical matrix diagonalization) and mode oscillator strengths. The latter quantifies mode contributions to the frequency-dependent permittivity. The optimization leads to Bloch skyrmion as presented in Fig. S5c and in agreement with both experimental findings and calculations within the second-principles framework[11]. To understand the role of selected G-point phonon modes, we analyze their associated $\Delta P$ patterns, which contain the information about the modes' topology and symmetry.

For a more direct comparison of the experimentally evaluated x-ray dispersion of THz excited modes, we investigated a dynamical structure factor $S(Q_x, \omega)$ calculated from a molecular dynamics trajectory. To this end, the optimized $20 \times 20 \times (PbTiO_3)_{16}/(SrTiO_3)_8$ simulation box has been multiplied 16 times in the in-plane direction, x, leading to a structure containing 16 skyrmion bubbles in a row (this ensures high resolution in the corresponding direction of the reciprocal space). Molecular dynamics simulation was performed using DL_POLY3, setting the temperature at 10 K, low enough to ensure that the correspondence with the 0 K phonon calculation is maintained. After 100 ps of equilibration time in the NVT ensemble, 200 ps of NVE trajectory has been produced for further analysis. $S(Q_x, \omega)$ dispersion curves are then calculated using the program mp_tools[56].

## Data availability

The data supporting the findings of this study are reported in the main text and supplementary materials. Raw data are available from the Open Science Framework at https://doi.org/10.17605/OSF.IO/YAXGV.

## Code availability

The codes used to produce the results are available from the corresponding author upon reasonable request.

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

## Acknowledgments

This work is primarily supported by U.S. Department of Energy, Office of Science, Office of Basic Energy Sciences, under Award Number DE-SC-0012375 for XFEL experimental design, data collection, and data analysis. R.R. acknowledge support from the Office of Basic Energy Sciences, US Department of Energy (DE-AC02-05CH11231). S.D. acknowledges the Scheme for Transformational and Advanced Research in Sciences (MoE-STARS/STARS-2/2023-0048) and the Indian Institute of Science start-up grant for financial support. L.W.M. and R.R. also acknowledge partial support of the Army Research Office under the ETHOS MURI via cooperative agreement W911NF-21-2-0162 for the development of superlattice structures. D.C., T.Y. and L.Q.C. also acknowledge partial support as part of the Computational Materials Sciences Program funded by the U.S. Department of Energy, Office of Science, Basic Energy Sciences, under Award No. DE-SC0020145. Y.C. and H.Wen acknowledge the support for data reduction by U.S. Department of Energy, Office of Science, Office of Basic Energy Sciences, Materials Sciences and Engineering Division. MAPG, MP and JH were supported by the Czech Science Foundation (project no. 19-
28594X), MAPG acknowledge the European Union and the Czech Ministry of Education, Youth and Sports (Project: MSCA Fellowship CZ FZU I —CZ.02.01.01/00/22_010/0002906). Computational resources for atomistic simulations were provided by the e-INFRA CZ project (ID:90254), supported by the Ministry of Education, Youth and Sports of the Czech Republic. Use of the Linac Coherent Light Source (LCLS), SLAC National Accelerator Laboratory, is supported by the U.S. Department of Energy, Office of Science, Office of Basic Energy Sciences under Contract No. DE-AC02-665 76SF00515. This research used in part resources of the Advanced Photon Source, a U.S. Department of Energy (DOE) Office of Science User Facility operated for the DOE Office of Science by Argonne National Laboratory under Contract No. DE-AC02-06CH11357, with data collected at 7ID-C and 33ID-B beamlines at the Advanced Photon Source (APS).

## Author contributions

V.A.S., H. Wang, A.M., Y.C., H.P., D.Z., T.S., S.S., M.C.H., P.K., S.N., Y.S., Q.N., A.L., J.F., V.G., H.Wen performed the measurements at the LCLS. C.D., T.Y., U.S. and L.-Q.C. performed the dynamic phase-field simulation. M.P., M.G., J.K., and J.H. performed the atomistic modeling. S.D., M.M., R.R., and L.W.M. synthesized the samples. V.A.S., Z.Z., J.F. and H.Wen performed the measurements at the APS. H.Wang, V.A.S. and H. Wen analyzed the data. H. Wang and H. Wen wrote the manuscript with inputs from all authors. H. Wen conceived the project. The work was supervised by J.F., J.H., V.G. and H.Wen.

## Competing interests

The authors declare no competing interests.

## Additional information

¹Department of Materials Science and Engineering, The Pennsylvania State University, University Park, PA, USA. ²Stanford Institute for Materials and Energy Sciences, SLAC National Accelerator Laboratory, Menlo Park, CA, USA. ³Advanced Photon Source, Argonne National Laboratory, Lemont, IL, USA. ⁴Institute of Physics of the Czech Academy of Sciences, Prague, Czech Republic. ⁵Materials Research Centre, Indian Institute of Science, Bangalore, India. ⁶Institut Laue Langevin, 71 avenue des Martyrs, Grenoble, France. ⁷Department of Materials Science and Engineering, University of California, Berkeley, Berkeley, CA, USA. ⁸Department of Materials Science and Engineering, Stanford University, Stanford, CA, USA. ⁹Materials Science Division, Argonne National Laboratory, Lemont, IL, USA. ¹⁰Linac Coherent Light Source, SLAC National Accelerator Laboratory, Menlo Park, CA, USA. ¹¹Materials Sciences Division, Lawrence Berkeley National Laboratory, Berkeley, CA, USA. ¹²Department of Physics, University of California, Berkeley, Berkeley, CA, USA. ¹³Department of Materials Science and NanoEngineering, Rice University, Houston, TX, USA. ¹⁴Department of Physics and Astronomy, Rice University, Houston, TX, USA. ¹⁵Rice Advanced Materials Institute, Rice University, Houston, TX, USA. ¹⁶Department of Chemistry, Rice University, Houston, TX, USA. ¹⁷Present address: Interdisciplinary Research Center, School of Mechanical Engineering, Shanghai Jiao Tong University, Shanghai, China. ¹⁸These authors contributed equally: Huaiyu Hugo Wang, Vladimir A. Stoica, Cheng Dai, and Marek Paściak. ✉e-mail: freeland@anl.gov; hlinka@fzu.cz; vxg8@psu.edu; wen@anl.gov

