## [Transparent Peer Review file · Nature Communications]

Terahertz-field activation of polar skyrons

Corresponding Author: Dr Haidan Wen

Version 0:

Reviewer comments:

Reviewer #1

(Remarks to the Author)

The manuscript entitled as "Ultrafast excitation of polar skyrons" by Huaiyu (Hugo) Wang et al. presents the study of polar skyrmions by revealing their nonequilibrium collective dynamics, which had remained largely unexplored or limited to theory or steady state experimental conditions.

Through the use of THz-pumped femtosecond x-ray diffraction, the authors directly observe the excitation and dispersion of collective modes of skyrons in a regime that conventional techniques had not accessed. A particularly striking discovery is the avoided crossing of the acoustic phonon band, which provides clear evidence of hybridization between phononic and polar topological modes. This phenomenon indicates that polar skyrmions fundamentally alter the transport of energy at the nanoscale. The selective excitation by THz fields is critical, as it couples specifically to ionic displacements rather than inducing electronic transitions. This allows for a cleaner observation of lattice responses and avoids complications such as electronic heating or carrier screening.

In addition, the manuscript presents the first direct evidence of ultrafast skyrmion dynamics, uncovering essential physical characteristics through dispersion relations and their dependencies on temperature and external bias. These findings offer new opportunities for controlling topological polarization textures in ferroelectric materials and hold significant promise for applications in ultrafast, high-density, and low-power information technologies.

Therefore, I recommend this manuscript for publication in Nature Communications after addressing the minor revisions.

1) In Fig. 3a, the comparison between experiment and simulation is not clearly presented. For effective comparison, the authors should consider a more efficient visualization method.

2) The origin of the peak structure for the $m=2$ mode near 330 K is unclear. The authors are encouraged to provide further discussion or clarification on this point.

Reviewer #2

(Remarks to the Author)

In their manuscript, the authors present an ultrafast excitation of collective polarization modes in polar skyrmions by a THz-pump and x-ray-probe experiment. The collective excitations are probed by a periodic intensity change in the vicinity of the skyrmion superlattice peaks that appear around the structural Bragg peaks of the lattice. The frequency of the oscillation as function of the scattering vector yields the dispersion relation that shows two modes and an avoided crossing that indicates the strong coupling of polarization and elastic waves. The opposite phase of the oscillation for different diffraction orders was related to the domain wall of the polar skyrmions via simulations that reproduced the main experimental observations.

Furthermore, the authors demonstrate the control of the amplitude of the collective excitations by transforming bubble domains to stripe domains either via electric-field or temperature.

I think the results have the potential of a significant impact in the field due to their very clear experimental results and the high quality modelling that nicely match the experimental results demonstrating a high degree of understanding that advance the field and might be interesting for a broader audience. I like the approach of a scattering probing technique tacking advantage of the quasi-four-fold symmetry of the polar skyrmions in the sample plane. And I want to highlight that the manuscript is a very good example how modelling can be used to understand complex experimental results in detail. The modelling and the systematic variation of the experimental conditions are very convincing.

However, I think the current version of the manuscript lacks a comprehensive discussion of the physical picture that will hinder the recognition of this work by a broader audience. After reading the manuscript several times, I have a hypothesis about the series of events and the critical mechanisms. But this should not be the goal of the manuscript. In contrast, either in

the main text or in the discussion there should be a paragraph dedicated to the physical picture the authors could achieve from their experiment and modelling about the mechanism and decisive properties, so that these insights are accessible to the reader.

Below, I listed some questions and comments that might improve this point when added to the current manuscript:

1. In line 40 on page 3, the modelling of the THz-induced dynamics is introduced for the first time. As the manuscript directly follows with learnings from the comparison of the modelling and the experiment, it would be useful, if the authors could add a short comment in which way the THz-induced perturbation is implemented. Is the THz-pulse assumed to drive a single phonon mode or an ensemble of (uncoupled) phonon modes? This information would provide a more physical picture of the proposed series of events by the modelling and will improve the understanding and trust of the reader.
2. In line 15 on page 4: What is meant with "modified energy transport different from the conventional transverse acoustic (TA) waves"? When I read "energy transport" I naively thought about the spatial distribution of energy. Do the authors mean an energy transfer among different degrees of freedom after the THz-deposition of energy to a certain mode?
3. Is there a reason why some measurements seem to be conducted in the vicinity of the 004 and some in the vicinity of the 013 Bragg peak (Fig. 1 vs. Fig. 3)? Do the authors expect differences in the skyron dynamics for different structural Bragg peaks?
4. Does the strong renormalization due to the polar skyrmions only affects a single TA phonon mode or are several modes involved? Maybe the simulation could be tuned to only allow for a single mode as perturbation and the amplitude of the modelled Oscillation could then identify the most dominant one.
5. Does the enhanced amplitude for the bubble domains mean that the DW is the most important feature for the coupling? If yes, the authors should explicitly state this in the mains text. If not, the authors should comment on the mechanism.
6. What is the reason for the vanishing avoided crossing during the transition from bubble to stripe domains?
7. Is the reduced THz-induced amplitude accompanied by an overall reduction of the SL peak intensity while heating? Could it be that the relative intensity modulation is the same for both types of domains?
8. Can the authors think of a mechanism to individually tune the amplitude of mode A and B? Maybe different THz frequencies to excite different phonon modes?

Some minor Remarks:

1. Extended Data Figure 1: In the title an "r" in Skyrmion is missing.
2. Is the legend in Extended Data Figure 1b correct? The phase of $m=-1$ and $m=-2$ is not opposite as in Fig. 2.
3. I find it hard to see the vanishing of the avoided crossing in Fig. 4b. Maybe the authors could guide the eye.
4. In Fig. 1b there is a question-mark box on top of the label of the color bar.

If the authors improve the discussion so that the physics behind their results are better accessible to the reader, I would recommend the publication of the manuscript in Nature Communications.

Version 1:

Reviewer comments:

Reviewer #1

(Remarks to the Author)

This manuscript presents the first observed ultrafast skyrmion dynamics.

And the authors have revised it well, resulting in a clearer presentation of both the text and the visual presentation of the figures, which effectively convey their findings.

Therefore, I recommend it for publication.

Reviewer #2

(Remarks to the Author)

The authors have taken my concerns into account and adapted their manuscript accordingly. I have no further remarks and think that the manuscript is now clear and presents novel physics with broad impact. Therefore, I recommend publication of the manuscript as it is in Nature Communications.

Response to Reviewer Comments

NCOMMS-25-40389-T: Ultrafast excitation of polar skyrons

Contents

- i. Response summary
- ii. Point-by-point response – Reviewer #1
- iii. Point-by-point response – Reviewer #2

Response summary

We sincerely thank the reviewers for their thorough reading of our manuscript and for providing constructive feedback. Per their suggestions, we have made changes to the main text, main figures, and supplementary information. These changes are highlighted in red in the updated manuscript and briefly summarized below:

1. Changed the title from “Ultrafast excitation of polar skyrons” to “Terahertz-field activation of polar skyrons” to highlight the central role of THz fields in exciting polar skyrons.
2. Added the following sentence in the abstract: “The key to active skyrons is the use of the THz field that couples primarily with skyrmion walls. ”
3. Added a sentence to clarify how THz field excitation was implemented in the phase-field simulation on page 3
4. Added panels f,g,h in Fig. 2 to illustrate the unique sensitivity to skyrmion domain walls at second order satellite peaks, as well as the discussion in the main text on page 4.
5. Revised Fig. 3a to improve visual comparison between experimental results and atomistic theory by providing high-resolution calculation results and optimizing the presentation.
6. Revised Fig. 4b to emphasize the reduced gap size in the labyrinth phase at 380 K, rather than a fully closed, linear dispersion.
7. Added a dedicated discussion on the physical picture of the process as requested by the reviewers on page 6.
8. Updated Extended Data Figs. 2, 3, 8, and 9 to better support the discussion, in response to the reviewer’s comments.
9. Added references 25, 27, 28, and resorted the reference list.
10. Cosmetic changes throughout, including grammatical corrections.

Reviewer #1 (Remarks to the Author, Author Responses, and Actions Taken):

Reviewer Comment: The manuscript entitled as “Ultrafast excitation of polar skyrons” by Huaiyu (Hugo) Wang et al. presents the study of polar skyrmions by revealing their nonequilibrium collective dynamics, which had remained largely unexplored or limited to theory or steady state experimental conditions.

Through the use of THz-pumped femtosecond x-ray diffraction, the authors directly observe the excitation and dispersion of collective modes of skyrons in a regime that conventional techniques had not accessed. A particularly striking discovery is the avoided crossing of the acoustic phonon band, which provides clear evidence of hybridization between phononic and polar topological modes. This phenomenon indicates that polar skyrmions fundamentally alter the transport of energy at the nanoscale. The selective excitation by THz fields is critical, as it couples specifically to ionic displacements rather than inducing electronic transitions. This allows for a cleaner observation of lattice responses and avoids complications such as electronic heating or carrier screening.

In addition, the manuscript presents the first direct evidence of ultrafast skyrmion dynamics, uncovering essential physical characteristics through dispersion relations and their dependencies on temperature and external bias. These findings offer new opportunities for controlling topological polarization textures in ferroelectric materials and hold significant promise for applications in ultrafast, high-density, and low-power information technologies. Therefore, I recommend this manuscript for publication in Nature Communications after addressing the minor revisions.

Author Response: We thank the reviewer for the accurate summary of our work and the recommendation for publication. We have addressed the comments as follows.

1) In Fig. 3a, the comparison between experiment and simulation is not clearly presented. For effective comparison, the authors should consider a more efficient visualization method.

Author Response: We examined Fig. 3a and found that the potential confusion may arise from the ambiguity about which part of the plot is experimental data, and which part is the simulation. We also think calculation with the enhanced frequency resolution will help improve the comparison. Therefore, we have recalculated the dispersion curve with higher energy resolution and optimized the color map so that the experimental and theoretical data are clearly visualized. In the figure caption, we explicitly indicate that the red dots are experimental data that are overlaid with the simulation.

Actions Taken: We have modified and improved Fig. 3a to facilitate better comparison between the experiment and simulation results (Fig. A1).

Fig. A1 Updated dispersion relation shown as Fig. 3a in the revised manuscript. Fourier spectra of the time evolution of the relative intensity change of Fig. 2a (left), compared with the results by the atomistic simulation (right). The color map of the simulation represents the amplitude of the dynamic structure factor $S(Q_x, \omega)$. Red dots are the overlaid experimental data with the error bars that show the fitting errors of the Lorentzian peak fitting of Fourier spectra at each Q_x . Two dispersion branches are labeled as A and B mode. The dispersion outside of the THz-pump spectrum shown on the right axis is not experimentally discernible. The dispersions of polariton (P), longitudinal/transverse acoustic (LA/TA) phonons are shown for comparison.

2) The origin of the peak structure for the $m=2$ mode near 330 K is unclear. The authors are encouraged to provide further discussion or clarification on this point.

Author Response: We thank the referee for this observation. We assume the reviewer refers to Fig. 4a. The amplitude of $m=2$ mode (observed at the $m=2$ satellite peak) reaches a maximum near 330 K while the amplitude of $m=1$ mode does not peak up at this temperature. Its origin can be traced to the different sensitivity of $m=2$ and $m=1$ peaks to the specific component of the skyrmion. As revealed by XRD simulations based on phase-field results, the second-order peaks exhibit enhanced sensitivity to skyrmion domain wall structures, while the first-order peaks are more sensitive to domain configurations (added Fig. 2f-h, Extended Data Fig. 3). The transition from skyrmion bubble to labyrinth (Fig. 4c) mainly involves the reconfiguration of domain walls. As the temperature approaches the critical transition temperature, the polarization of domain walls are more susceptible to the THz field and is easier to be driven coherently, compared with the domains. Therefore, the dynamical signal measured at the $m=2$ peak has a higher amplitude close to the transition temperature. In summary, the enhanced oscillation amplitude of the $m=2$ peak at 330 K is due to the enhanced coherent domain wall motion close to the transition temperature.

Actions Taken:

We have added a paragraph in the main text, under the section 'Control of Polar Skyrmions,' explaining the differing sensitivities of the $m=1$ and $m=2$ peaks and their resulting dynamic response behaviors near the critical transition temperature:

“The temperature-dependent amplitudes of the coherent oscillations at $m = \pm 2$ diffraction are different from those of the $m = \pm 1$ diffraction (Fig. 4c). The amplitude of the $m = \pm 2$ diffraction intensity oscillation reaches maximum at 330 K, while that of the $m = \pm 1$ diffraction does not, which reflects their distinct sensitivities to skyrmion structures. During the transition from skyrmion bubble to labyrinth structure, domain walls significantly reconfigure close to the transition temperature, leading to their enhanced susceptibility to the THz excitation and thus the increase of the oscillation amplitude at 330 K. Once the labyrinth structures dominate, domain walls density is significantly reduced, so as the dynamical responses. The correlation of reduced domain wall density and dynamical responses highlight the leading role of domain walls in driving the dynamic response.”

Reviewer #2 (Remarks to the Author, Author Responses, and Actions Taken):

Reviewer Comment: In their manuscript, the authors present an ultrafast excitation of collective polarization modes in polar skyrmions by a THz-pump and x-ray-probe experiment. The collective excitations are probed by a periodic intensity change in the vicinity of the skyrmion superlattice peaks that appear around the structural Bragg peaks of the lattice. The frequency of the oscillation as function of the scattering vector yields the dispersion relation that shows two modes and an avoided crossing that indicates the strong coupling of polarization and elastic waves. The opposite phase of the oscillation for different diffraction orders was related to the domain wall of the polar skyrmions via simulations that reproduced the main experimental observations.

Furthermore, the authors demonstrate the control of the amplitude of the collective excitations by transforming bubble domains to stripe domains either via electric-field or temperature.

I think the results have the potential of a significant impact in the field due to their very clear experimental results and the high quality modelling that nicely match the experimental results demonstrating a high degree of understanding that advance the field and might be interesting for a broader audience. I like the approach of a scattering probing technique tacking advantage of the quasi-four-fold symmetry of the polar skyrmions in the sample plane. And I want to highlight that the manuscript is a very good example how modelling can be used to understand complex experimental results in detail. The modelling and the systematic variation of the experimental conditions are very convincing.

However, I think the current version of the manuscript lacks a comprehensive discussion of the physical picture that will hinder the recognition of this work by a broader audience. After reading the manuscript several times, I have a hypothesis about the series of events and the critical mechanisms. But this should not be the goal of the manuscript. In contrast, either in the main text or in the discussion there should be a paragraph dedicated to the physical picture the authors could achieve from their experiment and modelling about the mechanism and decisive properties, so that these insights are accessible to the reader.

Author Response: We appreciate the reviewer’s constructive feedback to provide more discussions on the physical picture of the process in the main text. We have added a dedicated paragraph to further emphasize the underlying physical picture. The changes in response to the additional review comments below also help gain insights into the physics of this work.

Actions Taken: Besides other changes detailed for each comment below, we added a paragraph at the end of the discussion session to summarize the physical picture of our work: “The combined experimental and theoretical investigations clarify the overall physical picture of how a THz field activates polar skyrons – modes originating from the unique polarization configuration of polar

skyrmions. Although the THz field interacts with the dipole moment of every unit cell, the response at skyrmion walls is particularly strong, due to the polarization gradient leading to highly susceptible walls. The excitation of domain walls activates acoustic waves, which hybridize with the polarization waves. This hybridization, manifesting as a characteristic avoided crossing of phonon branches in the sub-THz range, results in the collective atomic motions in the form of dynamical swirling patterns.”

Below, I listed some questions and comments that might improve this point when added to the current manuscript:

1. In line 40 on page 3, the modelling of the THz-induced dynamics is introduced for the first time. As the manuscript directly follows with learnings from the comparison of the modelling and the experiment, it would be useful, if the authors could add a short comment in which way the THz-induced perturbation is implemented. Is the THz-pulse assumed to drive a single phonon mode or an ensemble of (uncoupled) phonon modes? This information would provide a more physical picture of the proposed series of events by the modelling and will improve the understanding and trust of the reader.

Author Response: Thanks for pointing this out. Although the implementation of the THz field was implicitly presented in the method section on phase-field simulation, we agree that it will be beneficial to explicitly show and discuss how the THz field drives the dynamical processes in polar skyrmions.

The THz-induced perturbation is implemented in the phase-field modeling by applying the measured THz waveform to the polar skyrmions. Mathematically, the time-dependent THz field $E(t)$ directly enters the free energy term in Eq. (2) as $F_{\text{electric}}(t) \propto E(t)P(t)$, where $P(t)$ is the polarization. More details can be found in the Method of Ref. [29] in the revised main text. As the THz pulse is a broadband pulse, it drives an ensemble of collective modes. The set of collective modes is excited within the spectrum window, as shown by the “pump spectrum” in Fig.3a. In our experiment, we didn’t see a signature of their coupling and can’t comment on whether these collective modes are coupled or not, which could be a good topic to study in future.

Actions Taken:

We have added the following sentence to the main text in page 3: “...the polar skyrmions structure is perturbed with the same THz waveform as in the experiment (see Methods).” In methods, we added “The time-dependent THz field $E(t)$ modifies the free energy term $F_{\text{electric}} \propto E(t)P(t)$. ”

2. In line 15 on page 4: What is meant with “modified energy transport different from the conventional transverse acoustic (TA) waves”? When I read “energy transport” I naively thought

about the spatial distribution of energy. Do the authors mean an energy transfer among different degrees of freedom after the THz-deposition of energy to a certain mode?

Author Response:

As the reviewer correctly pointed out, the energy transport we referred to here is the spatial distribution of energy, specifically, the energy transport via acoustic phonons. Since we only focus on the lattice degrees of freedom in this work, we do not intend to refer to the energy transfer among different degrees of freedom, but nanoscale mechanical energy transport across the sample through skyrmion bubbles. As shown in Fig. 3a, the dispersion of the TA branch is significantly modified. For example, the group velocity of the acoustic wave becomes slower, shown by smaller slope of the dispersion. These observations thus provide direct evidence that the transport of mechanical energy is modified.

Actions Taken:

We first added “at nanoscale” in the sentence to clarify we refer to the spatial distribution of energy. The revised sentence is now reads: “ ...modified energy transport at nanoscale via acoustic phonons, different from the conventional transverse acoustic (TA) waves. In addition, we have modified Extended Data Fig. 9c to better highlight the comparison between the linear dispersion observed in the 400 nm pump data and the hybridization gap opening in the THz pump data.

3. Is there a reason why some measurements seem to be conducted in the vicinity of the 004 and some in the vicinity of the 013 Bragg peak (Fig. 1 vs. Fig. 3)? Do the authors expect differences in the skyrmion dynamics for different structural Bragg peaks?

Author Response: We presented both 013 and 004 Bragg peak data to show the generality of the modified phonon dispersion. These modifications are independent of which Bragg peak is selected to probe. As shown in Extended Data Fig. 4, both atomistic simulations and experimental results reveal that modes A and B exhibit similar dispersion, despite some subtle differences that are expected due to the different sensitivity of the selected Bragg peaks to the atomic motion.

Action Taken: To clarify the motivation of showing both 013 and 004 Bragg peak, we have added a sentence in the main text on page 5 “Despite different Bragg peaks being probed, the time-domain responses and dispersions remain similar, demonstrating the generality of the modified phonon dispersion.”

4. Does the strong renormalization due to the polar skyrmions only affects a single TA phonon mode or are several modes involved? Maybe the simulation could be tuned to only allow for a single mode as perturbation and the amplitude of the modelled Oscillation could then identify the most dominant one.

Author Response: In the frequency range of interest, there are only acoustic modes present for the monodomain PbTiO_3 ; all the optic modes are well above 1 THz. Hence indeed, only the acoustic branches is renormalized. From the atomistic simulation at finite temperatures, we see strong renormalization of the TA phonon mode, while the LA mode presents significantly weaker coupling to skyron modes. This can be seen in the calculated dispersion around the $(8+Q_x, 0, 0)$ peak in which the LA dispersion is clearly presented (See Fig. A2a). The LA dispersion is less disruptive by the polar skyron modes compared with the modified TA dispersion (Fig. A2b).

Fig. A2 a, $S(Q, \omega)$ dispersion along the longitudinal direction $(8+Q_x, 0, 0)$ calculated in the same way as Fig. 3a and Extended data figure 4a,b&d. The most prominent feature is the LA dispersion that exhibits little modification. **b**, TA dispersion is modified more strongly than the LA dispersion (adapted from Fig. 3a).

The idea of single-mode excitation is indeed very interesting, however, rather difficult to realize within our current atomistic simulation machinery. To have a selective excitation we would have to run a simulation with an oscillatory field for the time long enough not to broaden the spectral range. This would effectively mean the real computational time of months, comparable to the one that we spent simulating the trajectory used for the calculation of $S(q, \omega)$ dispersion. Additionally, even at low temperatures we are not in the fully harmonic regime and lack of the energy transfer between the modes, even the relatively separated lowest-frequency ones, is by far not guaranteed. Realistically, one would have to first test these ideas with several simulations on a smaller system. This could be pursued in the future work.

Actions Taken: We added a sentence on page 4 of the manuscript to show the coupling with TA mode is stronger: “Both branch deviates from the linear TA rather than LA dispersion, indicating stronger renormalization with TA phonons.”

5. Does the enhanced amplitude for the bubble domains mean that the DW is the most important feature for the coupling? If yes, the authors should explicitly state this in the mains text. If not, the authors should comment on the mechanism.

Author Response: It is correct that the DW is important for the coupling of the skyrmion with the THz field. As shown by the phase-field calculation in Fig. A4, the $m=2$ satellite diffraction, for which the largest contribution is from the domain wall, responds strongly to the THz field compared to the first-order satellite. The experiment evidence of strong coupling between THz-fields and domain walls can also be seen in the comparison between the 400 nm pump and the THz pump data (Fig. 2a and Extended Data Fig. 9a). Strong excitation of the $m = 2$ peak was observed only upon the THz pump, while no clear dynamical changes of the $m = 2$ peak were observed upon the 400 nm pump. This contrast highlights the ability of the THz pulse to strongly couple to domain walls, whereas above-band-gap excitation primarily perturbs the domains through a stress induced by above-band-gap excitation, which predominantly contributes to the $m = 1$ peak.

To avoid confusion with the DW activation mechanism, we also removed the discussion about the displacive excitation mechanism in the “discussion” section.

Actions Taken:

We revised the title to “Terahertz-field Activation of Polar Skyrmions” to emphasize the role of THz pulses in triggering the dynamic response.

We added panel Fig. 2f-h (copied below as Fig. A3) to highlight the domain wall contribution to the second satellite diffraction peak.

We added the text to highlight the unique detection of domain wall contribution in the second order satellite peak: “Finally, a pronounced dynamical response of $m = \pm 2$ diffraction was observed only under THz excitation, but not discernible upon the 400 nm optical excitation. This highlights the THz pulse strongly couples to the domain walls, whereas optical excitation primarily couples to the domains but does not activate the domain walls.”

Fig. A3 Strong dynamical signal at domain wall (as in the revised Fig. 2) f Schematic of domain and domain wall regions in a polar skyrmion. g Simulation of the domain walls contribution to satellite diffraction. h Analytical simulation of dynamical responses that separate the contribution from domain and domain walls at the maximum polarization change (See Supplementary Text 1).

6. What is the reason for the vanishing avoided crossing during the transition from bubble to stripe domains?

Author Response: As the skyrmion bubbles transform to the stripe (we call them labyrinth) domains, the size of the bubble changes significantly. For example, along the extension direction of the skyrmion (Fig. A4), the acoustic wave can travel without the interruption by the domain walls thus more likely to follow the TA dispersion, as we experimentally observed (Fig.3b). But we agree that the avoided crossing does not completely vanish and partially remains at 380 K. This is consistent with the above interpretation that the stripe domain only extends the skyrmion bubble along one dimension but it is still confined along the orthogonal directions, therefore, the avoided crossing becomes weaker but not completely vanishing.

Fig. A4 Comparing the selected acoustic wave propagation (wavevectors shown by green arrows) in skyrmion and labyrinth phases.

Actions Taken: We added the following paragraph in the main text dedicated to the physical picture discussion of reduced avoided crossing in the labyrinth phase in the section “Control of polar skyrons”: “The weakened avoided crossing in the labyrinth phase can be attributed to changes in domain wall morphology. As skyrmion bubbles evolve into stripe-like labyrinth domains, their size increases along one dimension, allowing acoustic waves to propagate more freely in that direction and follow the TA dispersion. However, since the domains remain confined in the orthogonal directions, the avoided crossing is reduced but not fully eliminated.”

7. Is the reduced THZ-induced amplitude accompanied by an overall reduction of the SL peak intensity while heating? Could it be that the relative intensity modulation is the same for both types of domains?

Author Response: If we understand correctly, the concern here is that the reduced dynamical signal shown in Fig. 4a when the sample temperature increases is due to the reduction of static diffraction intensity, e.g., shown by Extended Data Fig. 8b. We point out the dynamical signal amplitude shown in Fig. 4a has been normalized by the static diffraction intensity I , i.e., $\frac{\Delta I}{I}$ as noted in the figure caption. Therefore, the relative intensity modulation is indeed reduced.

8. Can the authors think of a mechanism to individually tune the amplitude of mode A and B? Maybe different THz frequencies to excite different phonon modes?

Author Response: This is a great question. The individual tuning of mode A and B is possible if a narrowband (bandwidth < 0.05) and intense THz source is available. This type of THz sources is under development and could be interesting to apply to study our samples in future work.

Some minor Remarks:

1. Extended Data Figure 1: In the title an “r” in Skyrmion is missing.

Author Response: We added the missing “r” in the caption of Extended Data Figure 1.

2. Is the legend in Extended Data Figure 1b correct? The phase of $m=-1$ and $m=-2$ is not opposite as in Fig. 2.

Author Response: Thanks for pointing out this labelling error for $m=+2/-2$ curves. We corrected the legend in Extended Data Figure 1b. Now, the phases of $m=-1$ and $m=-2$ are opposite.

3. I find it hard to see the vanishing of the avoided crossing in Fig. 4b. Maybe the authors could guide the eye.

Author Response: We appreciate the careful reading our manuscript. After re-examining the 380 K data, we agree that the avoided crossing does not fully disappear; rather, the phonon gap is significantly reduced, as discussed in Fig. A3. To clarify this point, we have added guide lines in the updated Fig. 4.

4. In Fig. 1b there is a question-mark box on top of the label of the color bar.

Author Response: This may arise from mac-to-pc conversion. We have deleted the question-mark box in the revised manuscript.

If the authors improve the discussion so that the physics behind their results are better accessible to the reader, I would recommend the publication of the manuscript in Nature Communications.

Author Response: We thank the reviewer for the constructive comments which have helped to improve the discussion of the physics behind our results for the readers. We hope the reviewer finds our revised manuscript is ready for publication.